# Six-Year Prescription Pattern of Antimicrobial Use in Cats at the Veterinary Teaching Hospital of the University of Pisa

**DOI:** 10.3390/ani14030521

**Published:** 2024-02-05

**Authors:** Lucia De Marchi, Matilde Vernaccini, Valentina Meucci, Angela Briganti, Ilaria Lippi, Veronica Marchetti, Luigi Intorre

**Affiliations:** Veterinary Teaching Hospital, Department of Veterinary Sciences, University of Pisa, 56124 Pisa, Italy; lucia.demarchi@unipi.it (L.D.M.); matilde.vernaccini@phd.unipi.it (M.V.); angela.briganti@unipi.it (A.B.); ilaria.lippi@unipi.it (I.L.); veronica.marchetti@unipi.it (V.M.); luigi.intorre@unipi.it (L.I.)

**Keywords:** antimicrobial prescription, empirical associations, Highest Priority Critically Important Antimicrobials, prudent use, feline practice

## Abstract

**Simple Summary:**

This study retrospectively assessed six years of prescription patterns in cats at the Veterinary Teaching Hospital of the University of Pisa. Approximately 34% of all prescriptions included antimicrobials. Amoxicillin-clavulanic acid and enrofloxacin were the most frequently prescribed antimicrobials, primarily for genitourinary tract infections. Empirical associations accounted for 13.0%, with the combination of amoxicillin-clavulanic acid and enrofloxacin being predominant. The oral route was the main administration method. While prescriptions aligned with prudent use recommendations, antimicrobial susceptibility tests were infrequently conducted (5.2%), indicating a need for interventions to improve responsible antimicrobial use, and the implementation of stewardship programs, emphasizing diagnostics for future enhancement.

**Abstract:**

The use of antimicrobials has greatly contributed to improving animal health. However, their inappropriate use reduces their effectiveness in treating bacterial infections and contributes to the selection of resistance. This study aimed to retrospectively evaluate the six-year pattern (2017–2022) of antimicrobial use in cats visiting the Veterinary Teaching Hospital (VTH) of the University of Pisa (Italy). The total number of prescribed antimicrobials, the number of animals for which an antimicrobial was prescribed, and the total number of antimicrobial prescriptions showed a significant time trend decrease during the study period, except for the fixed-dose combinations. The most frequently prescribed antimicrobials were amoxicillin-clavulanic acid (Synulox) (39.1%) followed by enrofloxacin (29.8%). These antimicrobials were mostly prescribed to treat infections affecting the genitourinary tract (~30%), followed by the gastrointestinal tract, skin, and respiratory system affections. Antimicrobials in empirical associations represented 13.0% of the total antimicrobial prescriptions, and the combination of amoxicillin-clavulanic acid (Synulox) with enrofloxacin accounted for the majority. The oral route represented the main route of administration of prescribed antimicrobials, followed by parenteral and topical ones. Amoxicillin-clavulanic acid (Synulox) (37.2%), ceftriaxone (2.7%), and tobramycin (2.8%) were the most prescribed antimicrobials for the oral, parenteral, and topical routes, respectively. Antimicrobial prescriptions complied with prudent use recommendations in terms of availability of diagnosis, respect to the dose range, duration of treatment, and the use of medicinal products approved for the species. On the contrary, antimicrobial susceptibility tests were used infrequently (5.2%), lacking compliance with the existing guidelines observed in companion animal practice. Overall, additional interventions are required not only to improve the responsible use of antimicrobials in our feline practice but also to implement antimicrobial stewardship programs, enhancing diagnostics such as culture and sensitivity testing in the future.

## 1. Introduction

Knowledge of antimicrobial usage patterns is fundamental for implementing and monitoring antimicrobial stewardship (AMS) programs in veterinary practice, which generally refer to a series of interventions to monitor and direct antimicrobial use [1]. This knowledge represents one of the most effective ways to reduce antimicrobial resistance (AMR) [1,2,3,4,5]. A major contributor to AMR in veterinary practice is the use of antimicrobials in animal farming [6]. However, the presence of resistant bacteria in pets, which are likely to be transmitted to humans due to their close association and common environment, is also demonstrated [7,8,9,10,11,12,13]. Studies have shown that dogs and cats can carry multiple human-associated pathogens and a variety of multi-drug-resistant bacteria [14]. Consequently, antimicrobial use in companion animals is of increasing interest [15,16,17]. The problem of AMR has been extensively reviewed by the World Health Organization (WHO), which lists “critically important antimicrobials” into categories based on their importance in human medicine: Critically Important Antimicrobials (CIA), Highly Important Antimicrobials (HIA), and Important Antimicrobials (IA). Furthermore, a prioritization has been performed among CIAs to identify the Highest Priority Critically Important Antimicrobials (HPCIA). The HPCIA category includes quinolones, third- and higher-generation cephalosporins, macrolides and ketolides, glycopeptides, and polymyxins [1]. The Antimicrobial Advice Ad Hoc Expert Group (AMEG) suggested classifying antimicrobials into four categories (A to D—Avoid to Prudence) based on the availability of alternative antimicrobials in veterinary medicine within the European Union. This categorization aligns with the World Health Organization’s list of Critically Important Antimicrobials (CIAs) [18,19]. Recognizing the critical importance of certain antimicrobials and understanding their appropriate use in both human and animal medicine is essential for safeguarding public health, preventing the spread of resistant bacteria, and ensuring the continued efficacy of these antimicrobials in treating severe infections. The *One Health* perspective underscores the need for collaborative efforts between human and veterinary medicine, environmental science, and other relevant fields to address the complex challenge of antimicrobial resistance [20].

Recent data show that antimicrobial sales in Italy have decreased by 57.5%, reflecting a gradual commitment aimed at the continuous improvement of national performance [21]. Felines are one of the most represented populations, accounting for 10% of the total antibiotic sales volume. Among the antimicrobials intended for use in companion animals, the most prescribed are penicillins, first-generation cephalosporins, and macrolides [22]. Given that 10% to 25% of veterinary visits for pets globally result in the prescription of antimicrobials for cats, particularly for conditions such as urinary, skin, and respiratory diseases [23], it is crucial to consistently gather and analyze data on antimicrobial usage. This ongoing effort is essential for identifying and implementing interventions to mitigate antimicrobial resistance in both human and animal health. For this reason, the present study aims to retrospectively describe the six-year pattern of antimicrobial use at the Veterinary Teaching Hospital (VTH) of the University of Pisa (Italy), providing valuable insights to inform changes in practice. Additionally, these data may support the ongoing surveillance of antimicrobial use in companion animals, a key component of AMS.

## 2. Materials and Methods

### 2.1. Data Collection

This study included data on cat treatment records from 2017 to 2022, retrospectively extrapolated from the Veterinary Teaching Hospital (VTH) management software of the University of Pisa (Italy).

The analysis included only patients during routine visits. Cats referred to surgery and intensive care units were not included in the study due to the prevalent use of antimicrobial prophylaxis. All medical records were manually reviewed, and the data extraction was provided in Microsoft Excel format. Each medical record comprised the animal identification number (ID number); date and reason for the clinical evaluation; signalment; weight of the animal upon admission; diagnosis by the affected body system; and antimicrobial prescriptions.

### 2.2. Total Antimicrobial Prescriptions, Diagnosis, and Antimicrobial Susceptibility Tests

*Prescription*: Each prescription included both the indication and the duration of the treatment, other than the route and frequency of the administration. Prescriptions lacking complete information, which accounted for 1.3% of the total, were excluded from the study.

*Antimicrobial prescription*: Antimicrobial prescription involves choices regarding antimicrobial selection, dosage, route, and duration, and it may include the prescription of one or more prescribed antimicrobials. Antimicrobials prescribed in association were defined as empirical combinations (representing the simultaneous administration of two or more medicinal products), or fixed-dose combinations (representing medicinal products containing two or more antimicrobials) [24]. Antimicrobials were described according to the class, the route of administration, and the condition treated. They were categorized based on the anatomical therapeutic chemical (ATC) classification system for veterinary medicinal products [25]. Fixed-dose combination medicines were categorized in a distinct class. Quinolones, third- and higher-generation cephalosporins, macrolides and ketolides, glycopeptides, and polymyxins were categorized as Highest Priority Critically Important Antimicrobials (HPCIAs) via the application of the WHO criteria [26].

*Diagnosis*: The availability of a diagnosis was considered when assessing patients undergoing antimicrobial therapy. Clinical diagnosis was based on the evaluation of a patient’s signs and symptoms, medical history, physical examination, and laboratory tests, other than the direct identification of the causative microorganism. Bacteriological diagnosis involves the identification of the specific organism causing the infection, through culture tests from a clinical specimen.

*Antimicrobial susceptibility tests (ASTs)*: The susceptibility of bacterial isolates was evaluated by broth microdilution tests, allowing for the determination of the minimum inhibitory concentration (MIC) [27].

### 2.3. Compliance with Prudent Use

Compliance with prudent use recommendations was evaluated for systemic antimicrobials using the following criteria from published guidelines on prudent use: availability of a diagnosis; availability of antimicrobial susceptibility tests (ASTs); use of a medicinal product approved for the species; and respect of the posology (dose range and duration of treatment) reported on the information leaflet [28,29,30,31]. A dosage ± 10% of the recommended dose was considered correct [32,33].

### 2.4. Data Recording and Statistical Analysis

The existence of trends over the entire study period and the frequency distributions across six years in (I) medical consultation and prescriptions and (II) use of the antimicrobial classes, were tested by the Cochrane–Armitage test for linear trends and Fisher’s extract, respectively. Fisher’s extract test was also used to compare (I) the class of prescribed HPCIAs and (II) the distribution of antimicrobial prescriptions through route of administration.

The level of significance (*p*) was set at <0.05. All the statistical analyses were performed using Prism GraphPad 7.0 (GraphPad Software Inc., San Diego, CA, USA).

## 3. Results

### 3.1. Total Prescriptions

A total of 1749 prescriptions were issued for 1262 visited cats. Within the total number of prescriptions, 595 were antimicrobial prescriptions, accounting for 675 prescribed antimicrobials (Table 1). The total number of prescriptions showed a constant trend during the entire study period except the year 2020 due to the COVID-19 crisis, which caused a ~65% decrease in hospital activity compared to 2019 (Table 1). The total number of antimicrobial prescriptions, the total number of prescribed antimicrobials, and the number of animals for which an antimicrobial was prescribed showed a significant time trend decrease during the study period (Cochrane–Armitage test) (Table 1).

A total of 25 different antimicrobials, belonging to 11 different classes were prescribed (Table 2; Appendix A). The most prescribed antimicrobials during the six year study were the fixed-dose combination amoxicillin-clavulanic acid (Synulox) (*n* = 264), which corresponded to 40% of the total antimicrobials prescribed, followed by enrofloxacin (*n* = 169; 25%), the fixed-dose combination spiramycin-metronidazole (Stomorgyl) (*n* = 53; 8%), doxycycline (*n* = 34; 5%), and cephalexin (*n* = 30; 4.4%) (Table 2; Appendix A). During the study period, most prescribed antimicrobial classes showed a significant time trend decrease, except for the fixed-dose combinations, which exhibited the opposite trend compared to the other antimicrobial classes (Cochrane–Armitage test) (Table 2; Appendix A).

In the period 2017–2022, antimicrobials were mainly prescribed to treat infections affecting the genitourinary tract (*n* = 193), followed by the gastrointestinal tract (*n* = 114), the skin (*n* = 72), and respiratory systems (*n* = 41), corresponding to 28.6%, 16.7%, 10.7%, and 6.1% of the total number of prescribed antimicrobials, respectively (Table 3).

While 86.9% of antimicrobials have been prescribed as monotherapy, antimicrobials in empirical associations represented 13.1% of the total number of antimicrobial prescriptions, without statistical differences among years (Cochrane–Armitage test). The majority (54%) of the empirical associations included the veterinary medicine Synulox (amoxicillin-clavulanic acid), associated with (1) enrofloxacin (17 out of 28 (60%) were related to genitourinary diseases) or (2) spiramycin-metronidazole (Stomorgyl) (6 out of 6 (100%) were related to musculoskeletal conditions) or (3) tobramycin (4 out of 4 (100%) were related to ophthalmic diseases) or (4) marbofloxacin (4 out of 4 (100%) were related to gastrointestinal diseases) (Table 4).

The oral route accounted for 86.8% of antimicrobials and represented the main route of administration of prescribed antimicrobials regardless of the years. In contrast, the parenteral and topical routes of administration were less common, comprising only 6% and 7% of the prescribed antimicrobials, respectively (Table 5; Appendix A). The fixed-dose combination amoxicillin-clavulanic acid (Synulox) and enrofloxacin were the most orally prescribed antimicrobials, while the third-generation cephalosporin ceftriaxone and the aminoglycoside tobramycin were the most prescribed antimicrobials for the parenteral and topical routes, respectively (Table 5; Appendix A).

A total of 34% (227 out of 675) of the total prescribed antimicrobials were HPCIAs with a decreasing trend of prescription over the study period. Within the HPCIAs, the predominant class was fluoroquinolones, accounting for 88.5% of the entire HPCIA prescriptions (Table 2).

### 3.2. Compliance with Prudent Use

Conformity of antimicrobial prescription with the basic principles of prudent use recommendations was observed in terms of availability of diagnosis, duration of treatment, and respect for the dose range, achieving rates of 80%, 100%, and 73%, respectively. Non-compliance with the dose range accounted for 120 out of 439 (27.3%) prescribed antimicrobials: 66 (55%) and 54 (45%) non-compliances were identified as overdose and underdose prescriptions, respectively. Ninety-six percent of antimicrobial prescriptions consisted of medicinal products approved for the species (Table 6). On the contrary, antimicrobials were prescribed without the support of antimicrobial susceptibility tests (ASTs), which were performed only in 5.2% of the total number of prescriptions and 9.4% of those involving HPCIAs (Table 6).

## 4. Discussion

This study, conducted at the Veterinary Teaching Hospital (VTH) of the University of Pisa over six years (2017–2022), examined antimicrobial use in cats, providing both quantitative and qualitative insights to support continuous surveillance of their use, a crucial aspect of Antimicrobial Stewardship (AMS). The results showed a noteworthy decline over time in the total number of prescribed antimicrobials, the number of animals for which an antimicrobial was prescribed, and the total number of antimicrobial prescriptions. Our findings were consistent with reports from the European Medicines Agency (EMA), indicating a reduction in antimicrobial use in animals [21]. This alignment suggests that the initiatives and campaigns implemented by the European Union and national authorities may play a role in encouraging responsible antimicrobial usage. Moreover, frequent routine appointments, such as annual exams or other general practice activities that typically do not involve antimicrobial therapy, were three times more common than antimicrobial prescriptions. These results may justify the low rate of antimicrobial prescriptions in cats found in the present study.

In our study, approximately 30% of antimicrobials were prescribed for genitourinary tract infections, followed by gastrointestinal tract, skin, and respiratory system infections. Similarly, Aurich et al. [34] observed that bacterial urinary tract infection is a commonly diagnosed disorder in companion animals; another study showed that, in a survey of more than 3000 veterinary practitioners from 25 European countries, up to 62% of the antimicrobials were prescribed to treat genitourinary tract infections in cats [35]. For the treatment of this pathology in companion animals, HPCIAs, such as fluoroquinolones and third-generation cephalosporins, represent the first used choice [34,36,37,38,39]. Our results were partially in agreement with the cited studies, whereby third-generation cephalosporins were not frequently used, while approximately 30% of all prescribed antimicrobials were represented by fluoroquinolones. The latter constituted approximately 90% of the total number of prescribed HPCIAs. Fluoroquinolones are an antimicrobial class currently under scrutiny in veterinary medicine. Enrofloxacin is approved for use in dogs and cats [40], potentially justifying its high usage in the present study. However, even in situations where there is an authorized veterinary antibiotic that is a CIA, the off-label use of an unauthorized non-CIA product can be encouraged and considered prudent, provided it is supported by the best practices of evidence-based medicine and follows the scientific guidance on the responsible use of antimicrobials [41]. A study by Joosten et al. [42], investigating AMS and AMR in companion animals, reported that broad-spectrum antimicrobials and CIA antimicrobials represented 83% and 71% of the total number of treatments given to dogs and cats, respectively. Considering these concerns, it is crucial to comprehend veterinarians’ attitudes towards antimicrobial usage and the perception of the possible contribution of inappropriate use in the emergence of AMS. This understanding will facilitate better guidance for controlling the development of AMR. Beyond fluoroquinolones, the association amoxicillin/clavulanate is Europe’s most commonly prescribed antimicrobial [43]. Our results showed that ~40% of all prescribed antimicrobials were represented by fixed-dose combinations of amoxicillin-clavulanic acid, and ~11% of them were used to treat genitourinary tract infections. Amoxicillin-clavulanic acid is commonly used in dogs and cats to treat susceptible infections, which may include urinary, respiratory, or skin infections, due to their broad spectrum of action against Gram-positive and Gram-negative bacteria [44]. This provides a rationale for the outcomes observed in our study.

The oral route, despite its potential impact on AMR risk [45], constituted approximately 90% of administrations. Antimicrobial preparations are most frequently administered by the oral route in both dogs and cats, [33,39,46] since the owners of companion animals are often more willing to accept the oral administration of antimicrobial therapy instead of parenteral. Moreover, most of the authors studying antimicrobials in pets [33,34,39,46,47] concluded that amoxicillin-clavulanic acid was by far the most frequently used systemic antimicrobial by oral route, confirming the results observed in our study. Similarly, the examination of cat owners in the article by Cazer et al. [48] highlights that the owners show a preference for antimicrobials that are not only more cost-effective but also easier to administer [49]. The owners commonly sought specific antimicrobial formulations, especially liquids, based on their ease of administration. These findings underscore an opportunity for enhanced stewardship and communication between veterinarians and cat owners. The present study addressed concerns about empirical combinations, emphasizing potential adverse effects, pharmacological antagonism, and selection of resistant organisms [37,50]. While empirical combinations are common among veterinarians globally, our study, however, exhibited only 13% of empirical combinations on the total number of prescribed antimicrobials, counting amoxicillin-clavulanic acid with fluoroquinolones for the majority of the total empirical associations. Accordingly, the prescription of empirical combinations has been associated with the frequent use of broad-spectrum antimicrobials, such as amoxicillin with clavulanic acid and fluoroquinolones [51]. This is likely due to the ‘quick fix’ desire of pet owners following veterinarian-prescribed medication [47].

Our results showed that the majority of antimicrobial prescriptions complied with the principles of prudent use, in terms of availability of diagnosis. This means that antimicrobial therapy was administered to the majority of diagnosed patients. Additionally, the dose range, duration of treatment, and medicinal products approved for the species were respected. On the contrary, ASTs were infrequently used. ASTs, providing information on bacterial susceptibility, constitute a key element of antimicrobial stewardship programs. Thy enable the formulation of evidence-based recommendations to advocate for responsible antimicrobial use. Nevertheless, various studies suggest that, despite their significance, ASTs are often employed infrequently and typically only after the failure of the initial empirical therapy [23,33]. The primary obstacle to increased diagnostic testing in small animal practice, particularly ASTs, is frequently attributed to its cost [52,53,54,55,56]. This financial barrier has downstream effects, with high-cost cultures potentially biasing antibiograms (summaries of bacterial susceptibilities) and resulting in inappropriate choices of empiric antimicrobials. Furthermore, sample errors and the limited availability of interpretative criteria specific for species or organisms, on some occasions, do not allow the result to be interpreted and categorized with reasonable certainty and provide the clinician with a reliable interpretation. Numerous challenges persist in the implementation of AMS policies in feline medicine. Veterinarians face limitations in their awareness of antimicrobial stewardship guidelines, which have been consistently recognized as a hindrance in veterinary medicine [53,54]. Additionally, there is a lack of knowledge concerning AMS and AMR. Overall, this not only contributes to the lack of compliance with existing guidelines but also contradicts the fundamental principles of prudent use observed in companion animal practice.

## 5. Conclusions

This study offers valuable insights to support the ongoing monitoring of antimicrobial use in companion animals, serving as a crucial component of AMS. However, it was conducted exclusively at the Veterinary Teaching Hospital (VTH) of the University of Pisa, potentially limiting the generalizability of findings to other veterinary practices. The study sheds light on veterinary antimicrobial prescribing practices and the decreasing trend of antimicrobial use, but lacks an extensive exploration of pet owners’ motivations and behaviors, which could enhance understanding of the dynamics influencing antimicrobial usage. Addressing these limitations would improve the robustness and applicability of future studies in this field. Moreover, the future objective is to assist veterinary professionals in implementing the core principles proposed by UMN [57], including committing to stewardship, preventing common diseases through client education, judicious use of antimicrobial drugs, and evaluating use practices. Additionally, educating and building expertise, particularly in safe medication disposal, will be emphasized.

Establishing a surveillance system for proper antimicrobial utilization is crucial in reducing resistance occurrences, mitigating the emergence of multi-drug-resistant organisms, and enhancing patient care.

## Figures and Tables

**Table 1 animals-14-00521-t001:** Medical consultations and prescriptions from 2017 to 2022. Percentages were calculated over six years. ** *p* < 0.01; *** *p* < 0.001; **** *p* < 0.0001.

	2017	2018	2019	2020	2021	2022	Total 6 y	*p*-Value Trend
	*n* (%)	*n* (%)	*n* (%)	*n* (%)	*n* (%)	*n* (%)	*n* (%)	
Visited cats	267	228	238	108	188	233	1262	0.22
	(21.2)	(18.1)	(18.9)	(8.6)	(14.9)	(18.5)		
Total prescriptions	319	374	391	131	238	296	1749	0.11
	(18.2)	(21.4)	(22.4)	(7.5) **	(13.6)	(16.9)		
Antimicrobial prescriptions	150	219	83	25	47	71	595	<0.0001
	(25.2)	(35.8)	(14.0) ***	(5.7)	(7.9)	(12.0)		
Prescribed antimicrobials	167	256	92	29	54	77	675	<0.0001
	(24.7)	(37.9)	(13.6) ****	(4.3)	(8.0)	(11.4)		
Cats with at least one antimicrobial prescription	131	144	64	23	46	69	477	<0.001
	(27.5)	(30.2)	(13.4)	(4.8)	(9.6)	(14.5)		

**Table 2 animals-14-00521-t002:** Distribution of antimicrobial classes prescribed from 2017 to 2022. Percentages were calculated on the total number of prescribed antimicrobial classes per year. * *p* < 0.05; ** *p* < 0.01; *** *p* < 0.001; **** *p* < 0.0001.

	2017	2018	2019	2020	2021	2022	Total 6 y	*p*-Value Trend
	*n* (%)	*n* (%)	*n* (%)	*n* (%)	*n* (%)	*n* (%)	*n* (%)	
Aminoglycosides	6 (3.6)	11 (4.3)	5 (5.4)	1 (3.4)	2 (3.7)	1 (1.3)	26 (3.9)	<0.001
Cephalosporins	20 (11.9)	25 (9.8)	9 (9.7)	1 (3.4)	0 (0.0)	2 (2.6)	57 (8.4)	<0.0001
Fixed-dose combination	79 (47.3)	103 (40.2)	50 (54.3) *	13 (44.8)	27 (50.0)	51 (66.2) ****	323 (47.9)	<0.0001
Fluoroquinolones	51 (30.5)	83 (32.4) **	21 (22.8) ***	11 (37.9)	15 (27.8)	20 (26.0)	201 (29.8)	<0.0001
Fusidic Acid	1 (0.6)	2 (0.8)	-	-	-	-	3 (0.4)	<0.05
Lincosamides	2 (1.2)	6 (2.3)	1 (1.1)	-	1 (1.9)	1 (1.3)	11 (1.6)	<0.05
Nitroimidazoles	-	9 (3.5) ****	-	-	-	-	9 (1.3)	<0.01
Penicillins	1 (0.6)	-	-	-	-	-	1 (0.1)	0.11
Phenicols	1 (0.6)	1 (0.4)	-	-	-	-	2 (0.3)	0.07
Polymyxins	1 (0.6)	3 (1.2)	2 (2.2)	1 (3.4)	-	-	7 (1.0)	0.07
Tetracyclines	5 (3.0)	13 (5.1)	4 (4.3)	2 (6.9)	9 (16.7) *	2 (2.6) *	35 (5.2)	0.11
Total	167 (100)	256 (100) ***	92 (100) ***	29 (100) ****	54 (100) **	77 (100) *	675 (100)	<0.0001

**Table 3 animals-14-00521-t003:** Indications of the total number of prescribed antimicrobials by body system categories. Percentages were calculated on the total number of six-year prescribed antimicrobials.

	Cardiovascular	Gastrointestinal	Genitourinary	Lymphatic	Musculoskeletal	Ophthalmic	Otorhinolaryngology	Respiratory	Metabolic	Skin	No diagnosis	Total
Aminoglycosides	-	-	1 (0.1)	-	2 (0.3)	12 (1.8)	3 (0.4)	-	-	1 (0.1)	7 (1.0)	26 (3.9)
Cephalosporins	3 (0.4)	7 (1.1)	4 (0.6)	-	-	-	-	1 (0.1)	4 (0.6)	21 (3.1)	17 (2.5)	57 (8.4)
Fixed-dose combination	5 (0.7)	75 (11.1)	78 (11.6)	2 (0.3)	12 (1.8)	7 (1.0)	2 (0.3)	23 (3.4)	18 (2.7)	34 (5.0)	67 (9.9)	323 (47.9)
Fluoroquinolones	6 (0.9)	21 (3.1)	107 (15.9)	-	11 (1.6)	5 (0.7)	1 (0.1)	11 (1.6)	9 (1.3)	4 (0.6)	26 (3.9)	201 (29.8)
Fusidic Acid	-	-	-	-	-	-	-	-	-	3 (0.4)	-	3 (0.4)
Lincosamides	-	3 (0.4)	1 (0.1)	-	1 (0.1)	-	2 (0.3)	-	-	2 (0.3)	2 (0.3)	11 (1.6)
Nitroimidazoles	-	5 (0.7)	-	-	-	-	-	1 (0.1)	1 (0.1)	1 (0.1)	1 (0.1)	9 (1.3)
Penicillins	-	1 (0.1)	-	-	-	-	-	-	-	-	-	1 (0.1)
Phenicols	-	-	-	-	-	-	-	-	-	-	2 (0.3)	2 (0.3)
Polymyxins	-	-	-	-	-	-	6 (0.9)	-	-	1 (0.1)	-	7 (1.0)
Tetracyclines	1 (0.1)	2 (0.3)	2 (0.3)	2 (0.3)	-	5 (0.7)	-	5 (0.7)	4 (0.6)	5 (0.7)	9 (1.3)	35 (5.2)
Total	15 (2.2)	114 (16.9)	193 (28.6)	4 (0.6)	26 (3.9)	29 (4.3)	14 (2.1)	41 (6.1)	36 (5.3)	72 (10.7)	131 (19.4)	675 (100)

**Table 4 animals-14-00521-t004:** Empirical antimicrobial prescriptions in the total six-year period. Percentages were calculated on the total number of six-year prescriptions.

Antimicrobial in Empirical Associations	Prescriptions *n* (%)
II Associations	
Amoxicillin-clavulanic acid ^a^, Enrofloxacin	28 (35.9)
Amoxicillin-clavulanic acid ^a^, Spiramycin-metronidazole ^b^	6 (7.7)
Amoxicillin-clavulanic acid ^a^, Tobramycin	4 (5.1)
Amoxicillin-clavulanic acid ^a^, Marbofloxacin	4 (5.1)
Others *	36 (46.1)
Total	78 (100)

^a^ Synulox; ^b^ Stomorgyl; * Associations with the number of prescriptions < 4.

**Table 5 animals-14-00521-t005:** Antimicrobial prescriptions by route in the whole six year period. Percentages were calculated on the total number of six-year prescriptions. **** *p* < 0.0001.

	Route of Administration		
	Oral	Parenteral	Topical
	*n* (%)	*n* (%)	*n* (%)
Aminoglycosides	1 (0.1)	-	25 (3.7)
Cephalosporins	38 (5.6)	19 (2.8)	0 (0.0)
Fixed-dose combinations	306 (45.3) ****	12 (1.8)	5 (0.7)
Fluoroquinolones	186 (27.4) ****	10 (1.5)	5 (0.7)
Lincosamides	11 (1.6)	-	-
Penicillins	1 (0.1)	-	-
Phenicols	-	-	2 (0.3)
Polymyxins	-	-	7 (1.0)
Polymyxin B	-	-	7 (1.0)
Nitroimidazoles	9 (1.3)	-	0 (0.0)
Tetracyclines	36 (5.3)	-	1 (0.1)
Others	-	-	3 (0.4)
Total	586 (86.8) ****	41 (6.1)	48 (7.1)

**Table 6 animals-14-00521-t006:** Assessment of compliance for prudent use of antimicrobials from 2017 to 2022.

Prudente Use Recommendation	In Accordance *n* (%)
**Availability of:**	
- diagnosis ^a^	474 (79.7)
- antimicrobial susceptibility tests ^a^	31 (5.2)
- antimicrobial susceptibility tests ^b^	26 (9.4)
**Agreement with:**	
- use of a product approved for species ^c^	645 (95.5)
- respect of the dose range	319 (72.7)
- respect of the duration of treatment	410 (99.8)

^a^ On the total number of antimicrobial prescriptions; ^b^ On prescribed HPCIAs; ^c^ Off-label use: veterinary products *n* = 8 (1.8%); human products *n* = 22 (3.6%).

## Data Availability

Data are contained within the article.

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
