# Peer review of "Six-Year Prescription Pattern of Antimicrobial Use in Cats at the Veterinary Teaching Hospital of the University of Pisa"

_animals, 2024, doi:10.3390/ani14030521_

Round 1

Reviewer 1 Report

Comments and Suggestions for Authors

This study retrospectively evaluated a 6-year (2017-2022) pattern of antimicrobial use in cats at a veterinary teaching hospital, showing that antimicrobial prescriptions accounted for approximately 34% of total prescriptions, with a decreasing trend. Amoxicillin-clavulanic acid and enrofloxacin were the most commonly prescribed antimicrobials for various infections, mainly urogenital. Empirical combinations, especially amoxicillin-clavulanic acid with enrofloxacin, accounted for 13% of prescriptions. Oral administration predominated. While prescriptions generally adhered to prudent use recommendations, antimicrobial susceptibility testing was rarely performed (5.2%), highlighting the need for interventions to promote responsible use and implement stewardship programs to combat antimicrobial resistance.

General comments:

The reviewer commends for the valuable contribution of this paper. Such contributions are essential for advancing our understanding and addressing critical public health issues. Please check the full manuscript, especially the results section, when referring to the decrease trends.

Specific comments

Introduction

·       Line 50: [1,2-5]

Materials and Methods

·       Lines 77-78: “The analysis included only patients under ordinary visits, while cats referred to surgery and intensive care units were not included in the study” Why did the authors decide to exclude these groups? Even if they were referred to the VTM, this information could provide a valuable information on patterns of use. On the other hand, these locations are likely to be the main areas of AMRs in veterinary and human hospitals.

·       Lines 85-86: Did the authors have an estimate of the percentage of prescriptions missing complete information?

·       Line 102: Data Recording and Statistical Analysis

·        Line 109: Prism GraphPad 7.0 (GraphPad Software Inc., San Diego, California)?

·       Lines 127-128: The authors found that the most commonly prescribed classes of antimicrobial showed a significant decrease over time. However, the fixed-dose combination amoxicillin-clavulanic acid (Synulox) which constituted the 39.1% of the antimicrobials prescribed for cats in this 6-year period, showed an apparent increasing trend from 2018 to 2023, except for the interruption of the COVID-19, according to data presented in Table 2. Even if the p was significant, please check if the trend is decreasing or increasing. For example, in the case above (Amoxicillin-clavulanic acid) the trend increase.

Even if we consider the classes (instead of just this fixed-dose combination antibiotic), there is still an increase trend in the percentage of administration in this main group.

·       Tables 2 and 3: If the total prescriptions were 595 in the total 6-year period according to Table 2, then in Table 3, the reviewer would expect this same amount. However, authors presented a total of 675. Why this discrepancy between tables?

·       Table 6: Do the authors have the information on whether the non-respect of the dose range was mainly due to an overdose or a lower dose according to the established range of ± 10%?

Discussion

This section should be developed. Many results are not discussed, and the limitations of the study should be mentioned (exclusion of some cases, the fact that diagnosis doesn’t mean the need for antibiotics, etc.). Some of the statements are based on an incorrect interpretation of the significant results.

·       Line 192: [27,29-32]

·       Line 207: [25,32,38]

·       Line 209: [25,27,32, 39-39?]

·       Line 218:prudent use in terms of availability of diagnosis […]”. This approach is biased because, for example, the diagnosis of chronic diarrhea does not mean that the use of antibiotics is recommended. The authors should discuss these limitations.  

Author Response

The authors thank the reviewer#1 for the comments, enhancing the quality of the manuscript. All suggested revisions have been carefully considered, and the corresponding changes throughout the manuscript are highlighted in yellow.

Reviewer 2 Report

Comments and Suggestions for Authors

The study by De Marchi L. et al.  focuses on the use of antimicrobials in cats, admitted for ordinary visit in a teaching hospital during a 6-year period. In this paper various aspects linked to the antimicrobial prescription are mentioned. The data here proposed are the snapshot of the prescribed antimicrobials in a long period, in cats.

In this report I added some comments and suggestions to authors, to improve the description throughout the manuscript. I explained these comments and suggestions in more details below.

Introduction. The authors described the importance of the correct and prudent antimicrobial therapy in order to improve the antimicrobial stewardship (AMS) and to reduce the antimicrobial resistance (AMR).

Taking into account the role of the companion animal in the transmission of AMR (both via bacteria and genes), it is not explained the reason of choosing the cat among the pets for this study. Moreover, the authors cited data on sales (line 58) but they did not report data on the usage of antimicrobials in cats, representing the focal point of this paper. Other points are subsequently summarized.

Line 54-56. WHO list focuses on antimicrobials, ranking them into three major categories (critically important antimicrobial (CIA); highly important antimicrobial (HIA), and important antimicrobial (IA). Taking into account the different classes of antimicrobials comprised in this paper, the authors must cite all of them, explaining the meaning and their importance in a one health scenario, as well. Analogously, for EMA-AMEG categorization, a better explanation of the different categories for veterinary medicine, should help to have more detailed information regarding the topic.

Materials and methods. In this section, the authos must describe how the diagnosis has been performed and what test was used to determine the susceptibility pattern of the bacteria.

Lines 87-94.

Do the authors have the reference for the definition of empirical and fixed dose combinations? As an example, Sreedhar D et al, 2006 defines the combination products, also known as fixed dose drug combinations (FDCs), as the combinations of two or more active drugs in a single dosage form. The Food and Drug Administration, USA defines a combination product as ‘a product composed of any combination of a drug and a device or a biological product and a device or a drug and a biological product or a drug, device, and a biological product’.

Line 90-91: the authors must cite the reference for the ATC system. This system, used for the classification of veterinary medicines, is represented by an ATC code. It is not clear what the meaning of this proposed classification in this paper.   

The classification of the antimicrobials classes must be better described (lines 86-91).

Line 94. The authors cited the so called Highest Priority Critically Important Antimicrobials. They have to explain the role and the importance of these drugs for human beings, as well (see the introduction). The authors have to mention the full name followed by the acronym in parenthesis (Highest Priority Critically Important Antimicrobials (HP CIA).

Line 100. The authors cited as reference no. 23 the article from Weese, J.S. et al. regarding the guidelines for antimicrobial use in horse. In addition, the other two references are referred to the rational use of antimicrobials in animals/veterinary medicine, in general. The authors should consider that other guidelines are available for pet (e.g. the guideline for the use of antimicrobials in pet is available at the website of the italian Health Ministry).

Results. The authors reported that the overall prescription were 1749. Among them, only 595 belong to antimicrobial prescription: this is deducible from the table 1. The authors must report this data, in addition to the number of cats with at least one prescription. They must define in a clear manner, what number is considered for the calculation of the following percentages, as well (see table 2, e.g.; lines121-126).

Line 117. The authors described a significant decrease of the prescription in the studied period. The statistical analysis is not reported.

Table 2. the total of the prescribed antimicrobials is 675, differently from what reported in line 112 and in the table 1.

Line 127. See the comment for the line 117.

 Lines142-144. The authors must define how many prescription are represented by a unique antimicrobial (it means: what stands for vast majority?) vs the number of the empirical association.

Line 144: see the comment for line 117.

Lines 144-147. The authors should associate the diseases/diagnosis to the empirical associations.

Lines 168-175. The authors studied the compliance of the antimicrobial prescription in terms of availability of the diagnosis. This point is not described along the study: what the author intended as diagnosis for a bacteriological pathology?

Discussion.

According to the authors, a prudent and rational antibiotic therapy should be based on a susceptibility test of the isolates, responsible of the disease (prevously dignosed with a bacteriological test), as the authors stated in lines 96-98. Moreover, the use of the empirical therapy should be limited. This approach should be ground on the knowledge of the most involved bacteria in the pathogenesis, anamnestic information and epidemiological data on the isolates. These features represent the fundamental prerequisite for choosing the right antimicrobials (prudent use), but t is desirable performing an AST after 2-3 days in order to improve/correct the therapy.

In this paragrah these aspects must be better explained, based on the obtained results. In fact, the mentioned elements linked to a prudent use are not well discussed, taking into account that the AST was performed in a very low percentage, and that is not cited if a guideline has been followed. Moreover, another point to be discussed is represented by the use of high percentage of CIA and the use of oral administration, in a prudent use scenario, as well. Moreover, (lines 210-215), the authors must better explaine how this study had addressed concerns about empircal combinations in terms of emphasis of adverse effects, pharmacological antagonism, and selection of resistant organisms

Comments on the Quality of English Language

Line 81: please correct diagnoses with diagnosis

Line 84: authors should prefer indications (pl) vs an indication, for the medical use

Lines84-86. The authors should take into account to revise the phrase by omitting the colons. “Each prescription included both the indication and the duration of the treatment other than the route and frequency of the administration”

Author Response

The authors appreciate the comments of Reviewer#2 and are thankful for the suggestions, which have enhanced the scientific quality of the work. All comments have been carefully considered, and the corresponding modifications have been highlighted in yellow throughout the manuscript.

Reviewer 3 Report

Comments and Suggestions for Authors

I would like to see the author's be more assertive (in light of their findings) on how things could be done better in the future.

The tabulated data are hard to assimilate! Perhaps place some of the larger bocks of information in an appendix?

Author Response

Comments and Suggestions for Authors

I would like to see the author's be more assertive (in light of their findings) on how things could be done better in the future.

R: The discussion has been rewritten following the comments from reviewers and the editor, improving and clarifying important details. Additional information has also been included in the conclusion section, making the work more comprehensive and assertive

The tabulated data are hard to assimilate! Perhaps place some of the larger bocks of information in an appendix?

R: Tables were modified

Round 2

Reviewer 2 Report

Comments and Suggestions for Authors
  • The reviewer thanks the authors for the provided  answers to the different points. Both the entire article and the specific sections have been improved.